# A Method to Determine Human Skin Heat Capacity Using a Non-Invasive Calorimetric Sensor

**DOI:** 10.3390/s20123431

**Published:** 2020-06-17

**Authors:** Pedro Jesús Rodríguez de Rivera, Miriam Rodríguez de Rivera, Fabiola Socorro, Manuel Rodríguez de Rivera, Gustavo Marrero Callicó

**Affiliations:** 1Departamento de Física, Universidad de Las Palmas de Gran Canaria. E-35017 Las Palmas de Gran Canaria, Spain; pedrojrdrs@gmail.com (P.J.R.d.R.); miriam.mrdrs@gmail.com (M.R.d.R.); fabiola.socorro@ulpgc.com (F.S.); 2Instituto Universitario de Microelectrónica Aplicada (IUMA), Universidad de Las Palmas de Gran Canaria, E35017 Las Palmas de Gran Canaria, Spain; gustavo@iuma.ulpgc.es

**Keywords:** direct calorimetry, heat conduction calorimeters, human skin, medical calorimetry, thermal resistance, heat capacity

## Abstract

A calorimetric sensor has been designed to measure the heat flow dissipated by a 2 × 2 cm^2^ skin surface. In this work, a non-invasive method is proposed to determine the heat capacity and thermal conductance of the area of skin where the measurement is made. The method consists of programming a linear variation of the temperature of the sensor thermostat during its application to the skin. The sensor is modelled as a two-inputs and two-outputs system. The inputs are (1) the power dissipated by the skin and transmitted by conduction to the sensor, and (2) the power dissipated in the sensor thermostat to maintain the programmed temperature. The outputs are (1) the calorimetric signal and (2) the thermostat temperature. The proposed method consists of a sensor modelling that allows the heat capacity of the element where dissipation takes place (the skin) to be identified, and the transfer functions (TF) that link the inputs and outputs are constructed from its value. These TFs allow the determination of the heat flow dissipated by the surface of the human body as a function of the temperature of the sensor thermostat. Furthermore, as this variation in heat flow is linear, we define and determine an equivalent thermal resistance of the skin in the measured area. The method is validated with a simulation and with experimental measurements on the surface of the human body.

## 1. Introduction

The study of the thermal dissipation of the human body is of great interest in multiple fields. In air conditioning projects, it is necessary to know the dissipation of the occupants according to their activity. In the study of a subject’s metabolism, thermal dissipation is determined indirectly, measuring the absorbed VO2 or VCO2 emitted by the subject [1]. In the field of human physiology, all available data and techniques are used. Contact and remote thermometry are irreplaceable tools. There are numerous publications that provide temperature data of different areas and organs of the human body [2]. The thermal properties (thermal conductivity and specific heat capacity) of different parts of the human body are also of interest. Heat capacity is mainly determined from tissue sample analysis using techniques such as differential scanning calorimetry (DSC). These results are used for the study of pathologies [3,4] and the development of new bio-thermal models [5,6,7].

Sensors applied to the skin are of great importance as they are part of non-invasive diagnostic techniques [8,9]. New sensors applicable to the skin are currently being developed to determine its thermal conductivity [10] and also to determine the core temperature of the human body [11,12]. 

Currently, new sensors with materials adaptable to the skin are being developed [13,14,15,16]. This new generation of sensors perform measurements on the skin with a thermal penetration depth of 0.15 mm (stratum corneum). However, the sensor object of this work provides macroscopic thermal values with a thermal penetration depth of 3–4 mm. We believe these measurements with a higher thermal penetration depth are of interest and complement other thermal measurements.

This work introduces a new application of a calorimetric sensor that allows us to obtain the thermal properties of the skin. This calorimetric sensor has been developed to directly measure heat dissipation on localized surfaces of the human body. The working principle consists of applying the sensor to the skin (5–15 min) and determine the heat flow that is transmitted by conduction from the skin to the thermostat inside the device. Two prototypes with different configurations have been built, the first one with a measuring surface of 36 cm^2^ [17,18] and the second one with a measuring surface of 4 cm^2^ [19,20]. The second one, with smaller size, has a faster response and is easier to handle. With this second sensor, measurements have been performed on different subjects and its efficacy has been verified for measurements made with constant temperature of the sensor thermostat [21,22]. It has been experimentally found that for different constant thermostat temperatures, the heat flux decreases linearly as the thermostat temperature increases. This linear relationship allows us to define a thermal resistance of the skin in the measured area. This thermal resistance, in KW^−1^, is defined as the inverse slope of such line.

All measurements require a good starting and ending baseline. Therefore, these measurements have three stages: initial baseline (5 min), skin measurement (5 min), and final baseline (5 min). The first and third stages take place with the sensor placed on a calibration base. To determine the thermal resistance of the skin with acceptable precision, it is necessary to carry out at least four measurements, which implies a total time of one hour.

To reduce the application time of the sensor on the skin and thus obtain thermal results in a shorter measurement time, a new measurement method has been proposed. This new method consists of programming a linear variation of the thermostat temperature when the sensor is measuring on the skin. This method has two advantages: the first is that it allows us to determine in a single measurement the variation of the thermal dissipation of the skin as a function of the temperature of the thermostat, defining the thermal resistance of the skin. The second advantage is that it also allows us to determine the heat capacity of the measured area. 

This new measurement method also involves a new method of calculating the heat flow, which allows us to obtain the heat capacity of the dissipation zone. Once the heat capacity is known, the transfer functions that link the inputs to the outputs can be built for each measurement. On the other hand, variation in baselines is always a problem in thermal measurements. This variation is usually due to variations in the ambient temperature or the thermostat, although it may also be due to modifications of the experimental system during the measurement [23]. In our case there is a clear variation of the baseline due to the different ambient temperature that surrounds the sensor when it is placed at its base and when it is applied to the surface of the human body. The proposed method makes an evaluation of these variations and allows us to correct the baselines of each measurement. 

With respect to the medical applications of this sensor, we can find applications in the field of human physiology and in the detection of some pathologies that have a clear thermal component. This is currently the case with the digital thermography technique [24] that has allowed for the diagnosis of skin tumours [25] and monitoring of the evolution of different interventions such as knee replacements [26], inflammations [27], allergies [28], tendinitis [29], etc.

The research work is presented as follows: Firstly, the sensor and operating model are briefly described, based on a two-inputs, two-outputs system. The inputs are (1) the power dissipated by the skin and (2) the power dissipated in the sensor thermostat to keep the programmed temperature. The outputs are (1) the calorimetric signal and (2) the thermostat temperature. Once the mathematical model that relates the inputs to the outputs is established, the invariant parameters of the model are identified; this is done using Joule calibration measurements. The calculation method and the results of the experimental measurements carried out on the human body are presented below. Finally, the conclusions of this work are presented.

## 2. Materials and Methods

### 2.1. The Calorimetric Sensor

The experimental system has already been described in previous works [19,20], but in order to explain the calorimetric model proposed in this work, it is necessary to briefly describe the elements that constitute the sensor. The core of the sensor is a measurement thermopile (part 2 in Figure 1) located between an aluminium plate (part 1 in Figure 1) and a thermostat (part 3 in Figure 1). The thermostat is a 10 × 10 × 4 mm^3^ aluminium block that has an RTD (resistance temperature detector) and a heating resistance inside. The power dissipated in this resistance is determined by a PID controller (proportional–integral–derivative controller) that maintains the programmed temperature.

The thermostat has a cooling system based on a Peltier effect cooling thermopile, a heatsink and a fan (parts 4, 5 and 6 in Figure 1). The sensor assembly, excluding the heatsink-fan block, has a thermal insulation element (part 7 in Figure 1) to decrease external disturbances.

The purpose of the sensor is to measure the heat flux (*W*_1_) transmitted by conduction from the surface of the human body to the thermostat. As a consequence of this heat flow and due to the Sebeeck effect, the measurement thermopile provides the calorimetric signal (*y*). On the other hand, this heat flow will produce a modification of the power dissipated in the heating resistance (*W*_2_) that keeps the temperature of the thermostat under control (*T*_2_).

To calibrate the sensor, a base has been constructed consisting of a block of insulating material (part 8 in Figure 1) that has a copper plate that contains a resistance for Joule calibration (part 9 in Figure 1). In addition, the base has a magnetic clamping system (part 10 in Figure 1) that facilitates sensor handling. Sensor calibration is performed by dissipating known powers at the base while the temperature of the sensor thermostat is controlled. Calibration measurements are designed trying to reproduce the measurement conditions in the human body. The recalibration of the sensor is completed when the sensor is disassembled for technical reasons or when it is necessary to verify its operation.

The method of measurement in the human body is as follows: initially the sensor is located in the calibration base until the thermal equilibrium is reached for the programmed temperature of the thermostat (initial steady state). The time required is 10 min from when the system is turned on. It is then placed on the surface of the skin where the measure will take place. The time applied to the skin depends on the type of measurement to be performed. A measurement with the constant thermostat temperature to determine only the heat flow requires 5 min, but a measurement with a variable temperature of the thermostat to determine, in addition, the thermal properties of the skin, requires 12.5 min. Finally, the sensor is placed back in the calibration base. 

There are clear differences between calibration measurements and measurements on human skin, since in the first case the sensor is permanently located at the base, and in the measurements on human skin there is some movement of the sensor from the base to the surface of the human body. Such differences are important and will be tackled throughout this work.

### 2.2. Calorimetric Model

The device is modelled as a two-inputs, two-outputs system. The inputs are the power that passes through the sensor and that is to be measured (*W*_1_), and the power dissipated in the thermostat (*W*_2_) that maintains the programmed temperature of the thermostat. The outputs are the calorimetric signal (*y*) and the thermostat temperature (*T*_2_).

The signal-to-noise ratio of the output signals in an experimental measurement on human skin varies from 300 to 3000, (50 dB to 70 dB). This signal-to-noise ratio allows the identification of up to two poles of the transfer functions (TFs) that link the inputs to the outputs. For this reason, a simple two-element model is proposed (Figure 2). This modelling, called localized constants, is frequently used in heat conduction calorimeters [30,31,32]. The first element represents the domain where the heat dissipation to be measured takes place (*W*_1_). When the sensor is applied to the human body, this domain is the human skin. When the sensor is in the calibration base, this domain is the copper plate where the Joule calibration resistor is located. The second element represents the thermostat. The power dissipated in this domain (*W*_2_) is determined by a PID controller that holds the programmed temperature. The elements have *C*_1_ and *C*_2_ heat capacities. As the thermal conductivity of these elements is infinite, this implies considering a temperature in each domain (*T*_1_ and *T*_2_). These elements are thermally coupled to the outside and to each other by *P*_12_*, P*_1_ and *P*_2_ thermal conductance couplings. The outside is configured by the environment that is at a *T_room_* temperature, and the cooling system that is at a *T_cold_* temperature (see Figure 2). 

The power developed in each element (*W*_1_ and *W*_2_) is equal to the accumulated heat power in the domain plus the power transmitted by conduction to the neighbouring domains. The following equations (Equation (1)) describe the energy balance that relates the powers to the temperatures of each domain [19].
(1)$$\begin{array}{l} W_{1}(t)=C_{1}\frac{dT_{1}}{dt}+P_{1}\left(T_{1}-T_{room}\right)+P_{12}\left(T_{1}-T_{2}\right)\\ W_{2}(t)=C_{2}\frac{dT_{2}}{dt}+P_{2}\left(T_{2}-T_{cold}\right)+P_{12}\left(T_{2}-T_{1}\right) \end{array}$$

The calorimetric signal is a linear combination of the temperatures of the bodies that are in contact with each face of the measuring thermopile. That is, *y* = *k*(*T*_1_ − *T*_2_). 

Considering that the ambient temperature and the temperature of the cold focus (*T_room_* and *T_cold_*) are constant during the measurement and correcting the baselines of the variables (*T*_1_, *T*_2_, *W*_1_ and *W*_2_), we obtain an equivalent system of equations (Equation (2)) in which *T_room_* and *T_cold_* do not appear. In this case the values of the curves Δ*T*_1_, Δ*T*_2_, Δ*y,* Δ*W*_1_ and Δ*W*_2_ are zero at the initial instant [19].
(2)$$\begin{array}{l} \Delta W_{1}(t)=C_{1}\frac{d\Delta T_{1}}{dt}+P_{1}\cdot \Delta T_{1}+P_{12}\left(\Delta T_{1}-\Delta T_{2}\right)\\ \Delta W_{2}(t)=C_{2}\frac{d\Delta T_{2}}{dt}+P_{2}\cdot \Delta T_{2}+P_{12}\left(\Delta T_{2}-\Delta T_{1}\right);\;\Delta y=k\left(\Delta T_{1}-\Delta T_{2}\right) \end{array}$$

When the sensor is placed on its base, the ambient temperature *T_room_* remains practically constant. However, when the sensor is applied to the skin there are two significant changes. The first one refers to *C*_1_; now, this element is not the base but the area of the skin where the dissipation is being measured. This heat capacity is unknown and different in each area of the human body. The second change is related to the ambient room temperature around the sensor. This temperature in the neighbourhood of the skin is higher than the ambient temperature in the environment of the base. When the baselines of the experimental curves are corrected, the initial situation is corrected, eliminating the terms *T_room_* and *T_cold_* at the base, but the “new ambient temperature” in the neighbourhood of human skin is not taken into account. Therefore, we assume that the ambient temperature in the vicinity of the skin is higher by a Δ*T*_0_ value. 

We assume that this increase in temperature Δ*T*_0_ also affects the temperature of the cold focus, since *T_cold_* = *T_room_* + Δ*T_Peltier_*, where Δ*T_Peltier_* is the decrease in temperature produced by the cooling system. The value of Δ*T_Peltier_* depends on the supply voltage (*V_Peltier_*) of the thermopile and comes from Equation (3), whose parameters are determined experimentally [19,21].
(3)$$\Delta T_{Peltier}=a_{0}+a_{1}V_{Peltier}+a_{2}V_{Peltier}^{2}$$

In our case, *a*_0_ =1.7 K, *a*_1_ = −9.2 KV^−1^; *a*_2_ = 1.0 KV^−2^. In each measure *V_Peltier_* is constant and consequently Δ*T_Peltier_* is also constant. Thus, if there is a change in *T_cold_*, it is due only to changes in *T_room_*. When the sensor is placed on the skin, the outer surface of the thermal insulation maintains the initial temperature and gradually adapts to the new situation. This thermal adaptation of the surface is not instantaneous, and in the calculation, we will assume that this adaptation follows a variation given by (Equation (4)). The validity of this expression is checked by adjusting the experimental curves with the curves determined by the model.
(4)$$\Delta T_{0}=A\left(1-e^{-t/\tau_{T}}\right)+B\left(t/t_{end}\right)$$

The parameters *A*, *B* and the time constant *τ**_T_* are determined in each measurement. Time (*t*) begins when the sensor is applied to the skin and ends (*t_end_*) when the sensor is placed back on its base. The new equations (Equation (5)) incorporate this variation in the ambient temperature (Δ*T*_0_).
(5)$$\begin{array}{l} \Delta W_{1}(t)=C_{1}\frac{d\Delta T_{1}}{dt}+P_{1}\cdot \Delta T_{1}+P_{12}\left(\Delta T_{1}-\Delta T_{2}\right)-P_{1}\cdot \Delta T_{0}\\ \Delta W_{2}(t)=C_{2}\frac{d\Delta T_{2}}{dt}+P_{2}\cdot \Delta T_{2}+P_{12}\left(\Delta T_{2}-\Delta T_{1}\right)-P_{2}\cdot \Delta T_{0};\;\Delta y=k\left(\Delta T_{1}-\Delta T_{2}\right) \end{array}$$

By sorting this system of equations, and applying and operating the Laplace transform, six TFs (Equation (6)) are obtained that relate the inputs Δ*W*_1_(*s*), Δ*W*_2_(*s*) and Δ*T*_0_(*s*) with the outputs Δ*Y*(*s*) and Δ*T*_2_(*s*), where *s* is the Laplace variable. It is assumed that the initial value of all variables is zero and its derivative at the time origin is also zero and, for this reason, it is important to have a good initial baseline.
(6)$$\begin{array}{l} \left(\begin{array}{l} \Delta Y(s)\\ \Delta T_{2}(s) \end{array}\right)=\left(\begin{array}{lll} k\frac{C_{2}s+P_{2}}{Q(s)} & -k\frac{C_{1}s+P_{1}}{Q(s)} & \frac{k\left(P_{1}C_{2}-P_{2}C_{1}\right)s}{Q(s)}\\ \frac{P_{12}}{Q(s)} & \frac{C_{1}s+P_{12}+P_{1}}{Q(s)} & \frac{P_{2}(C_{1}s+P_{12}+P_{1})+P_{1}P_{12}}{Q(s)} \end{array}\right)\left(\begin{array}{l} \Delta W_{1}\\ \Delta W_{2}\\ \Delta T_{0} \end{array}\right)\;\\ \mathrm{being},\;Q(s)=\left(C_{1}s+P_{1}+P_{12}\right)\left(C_{2}s+P_{2}+P_{12}\right)-P_{12}^{2} \end{array}$$

As can be seen in (Equation (6)), the denominators of the six TFs are the same, which implies that the poles of all the TFs are the same. However, the numerators of each TF are different. It can be verified that the transient response of the calorimetric signal and the temperature of the thermostat depend on the heat capacity *C*_1_ where dissipation occurs. In general, an increase in *C*_1_ implies higher values of the time constants, which results in a slower transient response. Nevertheless, this will be discussed further in the simulation section.

### 2.3. Identification of the Model

The hypothesis proposed in the model implies that the parameters *C*_2_, *P*_1_, *P*_12_, *P*_2_ and *k* are characteristic of the sensor and invariant. On the other hand, the calorimetric signal (*y*), the temperature of the thermostat (*T*_2_) and the power dissipated in the thermostat (*W*_2_) are known. However, *C*_1_ depends on the area where dissipation *W*_1_ takes place. This power *W*_1_ is only known when performing calibration measurements: these measurements allow determining the invariant parameters of the model.

For calibration, the sensor is placed on its base and measurements similar to those made on human skin are programmed. Initially, a constant temperature of the thermostat *T*_20_ is programmed, and when the steady state is reached, a constant power *W*_10_ in the base is dissipated for 300 s. Then, for 150 s, a linear increase in the thermostat temperature is programmed until *T*_2*end*_ and simultaneously the dissipation in the base is linearly decreased from *W*_10_ to *W*_1*end*_. Finally, and for 300 s, the temperature of the thermostat *T*_2*end*_ is kept constant, also keeping the dissipation *W*_1*end*_ constant.

In Figure 3 the experimental curves of the input powers (*W*_1_ and *W*_2_) and the experimental curves of the output signals are represented: the calorimetric signal (*y*) and the thermostat temperature (*T*_2_). Two cases are shown: in the first one (green curves) the temperature increase in the thermostat is 10 K (from 26 to 36 °C); and in the second case (blue curves) the increase is 5 K (from 26 to 31 °C). In both cases the power *W*_1_ dissipated at the base (red curve) is 300 mW at the beginning and 100 mW at the end.

A minimization method based on the Nelder–Mead simplex algorithm [33,34,35] is used to identify the model parameters. The error criterion used is the sum of the roots of the root mean square errors (RMSE), weighted (Equation (7)). The root mean square error is calculated with the experimental curves (subscript *exp*) and those calculated by the model (subscript *cal*). 

*N* is the number of points and *α* is a weighting coefficient (*N* = 1800, *α* = 4). The calculated curves are determined with the system of equations (Equation (6)). The sampling period is 0.5 s.
(7)$$\varepsilon =\varepsilon_{y}+\alpha \cdot \varepsilon_{T_{2}}=\frac{1}{N}\sqrt{\sum_{i=1}^{N}{\left(y_{\exp}[i]-y_{cal}[i]\right)}^{2}}+\frac{\alpha }{N}\sqrt{\sum_{i=1}^{N}{\left(T_{2,\exp}[i]-T_{2,cal}[i]\right)}^{2}}$$

The calculation starts with some initial values of the model parameters (*C*_1_, *C*_2_, *P*_1_, *P*_2_, *P*_12_ and *k*). With these parameters the TFs that relate the inputs to the outputs (Equation (6)) are constructed, and the outputs calculated with the model (*y_cal_* and *T*_2*cal*_) are determined for the known inputs *W*_1_ and *W*_2_. Then the error criterion is applied (Equation (7)) and the Nelder–Mead method that provides new parameter values (*C*_1_, *C*_2_, *P*_1_, *P*_2_, *P*_12_ and *k*) is applied. The iterative process decreases the error (Equation (7)) until the fit between the calculated and experimental curves cannot be further improved. Figure 4 shows a diagram of the calculation procedure.

Table 1 shows the values of the parameters (*C*_1_, *C*_2_, *P*_1_, *P*_2_, *P*_12_, *k*) obtained in the calibration measurements. These results correspond to a series of 20 measurements in which the power dissipated in the base (400, 300 and 100 mW) and the increase in temperature of the thermostat (5 and 10 K) have been varied. The maximum RMSE in the calorimetric signal fittings is 14 µV over a 100mV peak-to-peak signal; and 4 mK on a 10 K peak-to-peak signal at the thermostat temperature fitting.

### 2.4. Simulation

Calibration of the sensor allows determining the model parameters that best fit the experimental measurements (Table 1). In this section the operation of the sensor is studied from simulations. (Equation (8)) shows the relationship between the inputs and outputs of the model by means of six transfer functions (*TF_i_*) whose expressions are indicated in the system of equations in (Equation (6)).
(8)$$\left(\begin{array}{l} \Delta Y(s)\\ \Delta T_{2}(s) \end{array}\right)=\left(\begin{array}{lll} TF_{1}(s) & TF_{2}(s) & TF_{3}(s)\\ TF_{4}(s) & TF_{5}(s) & TF_{6}(s) \end{array}\right)\left(\begin{array}{l} \Delta W_{1}\\ \Delta W_{2}\\ \Delta T_{0} \end{array}\right)$$

#### 2.4.1. Variation of the FT of the Model Depending on the Heat Capacity

A comparison of the TF is presented for different values of heat capacity. Figure 5 and Figure 6 represent the modulus of each *TF_i_* as a function of frequency, for extreme cases of heat capacity *C*_1_ = 3 JK^−1^ and *C*_1_ = 9 JK^−1^. The normalized module is represented to compare the dynamic responses with the same vertical scale.

In the calorimetric response, we observe that when *C*_1_ increases, the system responds slower, which is verified in *TF*_1_ and *TF*_3_; however, the dynamics of *TF*_2_ remains unchanged. On the other hand, the dynamic response of the thermostat temperature is less affected with the variation of *C*_1_, but as expected, the response of *TF*_4_ is slower with increasing values of *C*_1_. *TF*_5_ and *TF*_6_ are less altered with the variation of *C*_1_. 

#### 2.4.2. Simulations in the Calibration Base and in the Human Body

The simulations are performed from the systems of equations (Equation (5) or Equation (6)). First, we simulate the operation of the sensor when it is located in its base, which is completed for different heat capacities of the base: *C*_1_ = 3, 6 and 9 JK^−1^. The thermostat temperature is kept constant; when the sensor reaches steady state, it increases linearly to +10 K for 150 s and finally remains constant until the end. At the base, a constant power of *W*_1_ = 0.3 W is dissipated for 300 s, and when the thermostat temperature increases, it decreases linearly until *W*_1_ = 0.1 W, which remains constant until the end of the simulation. Figure 7 shows the thermostat temperature (*T*_2_), the calorimetric signals (*y*), the power dissipated in the base (*W*_1_) and the power dissipated in the thermostat (*W*_2_). Additionally, a 0.5 K linear increase in ambient temperature (Δ*T_0_*) has also been simulated. As expected, a higher value of the heat capacity *C*_1_ implies a slower transient response of the calorimetric signal.

Figure 8 shows the simulation of a measurement on the surface of the human body. The simulation takes into account that the sensor is initially at its base (*C*_1_ = 3 JK^−1^) and then it is applied to the skin, which has the same or different *C*_1_ heat capacity. When placing the sensor on the skin, three phenomena appear that make this measurement different from a calibration measurement at its base:

(1)There is an instantaneous contact between the sensor surface and the skin surface that are at different temperatures. This produces a peak in the calorimetric signal, caused by an instantaneous power that is transmitted from the highest temperature surface to the lowest temperature. This instantaneous power that is transmitted from the skin to the sensor is represented with an exponential function of the form
(9)$$W_{1}(t)=A_{0}+A_{1}\exp\left(-t/\tau_{1}\right)$$
where *A*_0_ is the power transmitted in steady state and *A*_1_ is the amplitude of the decreasing exponential due to instantaneous contact between the sensor surfaces and the skin. In the simulated case represented in Figure 8, we assume a power of *A*_0_ = 0.3 W for the initial temperature of the thermostat and a power of *A*_0_ = 0.1 W for the final temperature. We assume an amplitude of *A*_1_ = 2 W and a time constant of *τ*_1_ = 9 s.(2)The temperature surrounding the sensor has changed and is not the same temperature surrounding the sensor when it is in the base. In general, the temperature in the neighbourhood of the skin is higher. This temperature difference is represented by Δ*T*_0_ and responds to the expression given by (Equation (4)). In the simulated case *A* = 2.5 K, *B* = 1 K and a time constant of 9 s are considered.(3)The skin has a heat capacity typical of the area where it is being measured and therefore the transient response will depend on that heat capacity. Figure 8 shows these differences in the calorimetric signal. Three heat capacities for the skin have been used in the simulation: 3, 6 and 9 JK^−1^.

### 2.5. Method for Determining Heat Flux and Heat Capacity

This section describes the calculation method to determine the heat capacity and heat flow on a localized surface of the human body. The effectiveness of this method is verified from the curves simulated in the previous section. The following hypotheses are considered in the simulation of measurements on human skin:

(1)When the temperature of the thermostat is constant (*T*_2*initial*_) and the sensor is applied to the skin, the heat flow *W*_1_ that passes through the sensor obeys (Equation (9)). Contrastingly, when there is a linear variation in the temperature of the thermostat (from *T*_2*initial*_ to *T*_2*end*_), we assume that the power *W*_1_ decreases linearly to a final value, which remains constant while the temperature of the thermostat remains constant at its final value (*T*_2*end*_). Thus, the heat flux can be described with (Equation (10)):(10)$$\begin{array}{ll} W_{1}(t)=0 & for\;t<t_{1}\;\\ W_{1}(t)=A_{0}+A_{1}\exp\left(-(t-t_{1})/\tau_{1}\right) & for\;t_{1}\leq t\leq t_{2}\\ W_{1}(t)=W_{1}(t_{2})+\Delta A_{0}\cdot \left(t-t_{2}\right)/\left(t_{3}-t_{2}\right) & for\;t_{2}\leq t\leq t_{3}\\ W_{1}(t)=W_{1}(t_{3}) & for\;t_{3}\leq t\leq t_{end} \end{array}$$In this equation, *t*_1_ is the instant in which the sensor is applied to the skin, *t*_2_ is the instant in which the linear increase in the temperature of the thermostat begins, *t*_3_ is the instant in which the aforementioned linear variation ends and starts to keep the temperature constant, and *t_end_* is the final instant of the measurement.(2)The difference in ambient temperature Δ*T*_0_ obeys (Equation (4)).(3)The relationships between all the system variables obey the equations of the model (Equation (5)). The model parameters have been determined in the calibration (Table 1) except for the value of *C*_1_ which depends on the place of measurement.

The chosen calculation method is similar to that used in model identification and discussed in Section 2.3 of this work. The method consists of determining eight parameters: the first four (*A*_0_, *A*_1_, *τ*_1_, Δ*A*_0_) allow us to reconstruct the power *W*_1_(*t*). The next three (*A*, *B*, *τ_T_*) allow us to build the function Δ*T*_0_, and the last one (*C*_1_) completes the model described by (Equation (5)). 

The procedure begins with initial values of the first seven parameters with which the curves *W*_1_(*t*) (Equation (9)) and Δ*T*_0_(*t*) (Equation (4)) are constructed. From these temporal functions, the power *W*_2_(*t*) (known) and the initial value of the eighth parameter (*C*_1_), the temperature and calorimetric curve of the thermostat are reconstructed using the system of equations (Equation (5)). From these reconstructions, the error criterion given by Equation (7)) is determined. Using the Nelder–Mead simplex algorithm [33,34,35], the new parameter values are determined. Figure 9 shows a diagram of the calculation procedure.

Table 2 shows the results obtained for the three simulated measurements represented in Figure 8. The errors in the determination of the heat flow *A*_0_ and Δ*A*_0_ are less than 0.3%, and in the determination of the heat capacity it is less than 0.4%. The errors (RMSE) in adjusting the calorimetric signal and the thermostat temperature (less than 9 µV and 0.15 mK) are very small in relation to the peak-to-peak value of these curves (120 mV and 10 K). These results on the simulated measurements demonstrate that the proposed method is adequate.

## 3. Results 

In this section the results corresponding to measurements carried out on two healthy subjects are presented, the characteristic data of both subjects are indicated in Table 3. Both subjects are healthy, although the senior subject suffers from Hashimoto’s thyroiditis and is currently receiving treatment. 

### 3.1. Measurements in the Junior Subject

Once the suitability of the method in Section 2.5 has been verified, the proposed method is applied to an experimental measurement on human skin. We present a measurement made on the temple of the junior subject. The subject is sitting and resting during the measurement. Four identical consecutive measurements are made. Each of them is of the same type as those performed in the simulation. Figure 10 shows the experimental curves of the calorimetric signal (*y*), power dissipated in the thermostat (*W*_2_) and temperature of the thermostat (*T*_2_). Figure 11 shows the placement of the sensor on the subject’s temple. The measurements were made in the laboratory with an ambient temperature of 22.6 °C and a relative humidity of 53%. 

We apply the calculation method to each measurement. The results obtained are shown in Table 4 where the values of the eight parameters are indicated. Firstly, the parameters (*A*_0_, *A*_1_, *τ*_1_, Δ*A*_0_) related to the reconstruction of the heat flow through the sensor as a function of time when applied to human skin. Next, the value of the heat capacity of the area of the human body where the measurement is made is observed, the obtained average value being 5.91 JK^−1^. Finally, the parameters (*A*, *B*, *τ_T_*) that allow us to reconstruct Δ*T*_0_, which is the equalization of the temperature around the sensor when it is applied to human skin, are obtained. Adjustment errors are less than 35 µV for a 120 mV peak-to-peak calorimetric signal, and 4 mK for a 10 K peak-to-peak thermostat temperature.

Figure 12a shows the heat flow measured by the sensor in the first measurement made on the subject’s temple. The initial transient can be observed when applied from the sensor due to the instantaneous contact between the sensor surfaces and the skin. Next, the heat flow reaches a steady state of 288 mW for the thermostat temperature of 26 °C. This power decreases linearly as the thermostat temperature increases, reaching a value of 67 mW for the thermostat temperature of 36 °C. The reconstruction of the calorimetric signal and the thermostat temperature can be seen in Figure 12b, while Figure 12c displays the experimental and calculated curves indicated with the subscripts *exp* and *cal*, respectively. The variation Δ*T*_0_ as a function of time is also represented in Figure 12c. 

The power dissipated by the subject depends on the physical situation of the subject, the ambient, the temperature and the humidity, among other factors. We consider that a rigorous thermal characterization of the skin requires, in addition to knowledge of heat flow, data on the heat capacity and thermal conductance of the skin. The heat capacity and heat flux are determined directly by the calculation method. Thermal conductance (or inverse, thermal resistance), can be characterized from the variation of heat flow with the temperature of the thermostat. The inverse of the slope of this line has units of thermal resistance, therefore we define an equivalent thermal resistance of the skin *R_skin_* with the expression given by (Equation (11)).
(11)$$R_{skin}=\frac{\Delta T_{2}}{\Delta W_{1}}-R_{sensor}$$

The thermal resistance of the sensor *R_sensor_* is the inverse of the thermal conductivity *P*_12_. In this case, *R_sensor_ =* 1/*P*_12_ = 10.41 K/W, Δ*T*_2_ = 10 K, Δ*W*_1_ = ΔA_0_. Substituting the values from Table 4, we obtain *R_skin_* = 36.5 ± 1.7 K/W. The uncertainty of the values obtained for the thermal properties of the skin in this series of measurements is 3.5% for heat capacity and 4.7% for equivalent thermal resistance.

### 3.2. Measurements in Senior Subject

In this section, eight measurements performed on different days in the temple of the healthy 62 year-old male are presented. Table 5 shows the results obtained. Figure 13 shows the variation of the heat flow of each day as a function of time and the temperature of the thermostat. The heat flow is very similar on both days, although on the first day it is slightly higher. This could be because the ambient temperature is lower, which produces greater dissipation. As for the thermal properties of the skin in the measured area, the average heat capacity obtained on the first day is 5.7 J/K, on the second day it is 5.9 J/K, and the mean value is *C*_1_ = 5.8 ± 0.2 J/K. The average equivalent thermal resistance obtained on the first day is 31 K/W, on the second day it is 29 K/W, and the average value between the two days is 30 ± 2 K/W. The uncertainty in determining the thermal properties of the skin is 3.4% for heat capacity and 6.7% for thermal resistance. 

Table 6 shows results of two series of measurements performed on the right hand on two different days. In this case there, is a clear difference in the series of measurements from the first day given that the subject had an unusually low dissipation. The temperature around the sensor in the first day is *T_room_* + *A* = 21.3 + 0.61 = 21.91 °C, however, in the second day it is 23.68 °C. These values clearly show that the surface temperature of the skin is lower in the first case. On the other hand, we can compare the area *A*_1_*τ*_1_ which represents the over-energy of the transitory phenomenon. In the measurements of the first day, said energy is negative or very small compared to that of the second day. This is because the surface temperature of the skin before applying the sensor to the skin is lower on the first day. We also compared the heat capacities and equivalent thermal resistances obtained: on the first day *C*_1_ = 5.05 ± 0.20 JK^−1^, *R_skin_* = 38.9 ± 0.7 KW^−1^, and on the second day *C*_1_ = 5.43 ± 0.49 JK^−1^, *R_skin_* = 28.9 ± 1.3 KW^−1^. The heat capacities obtained in both measurements are less than in the temple and the values obtained in each day are of the same order of magnitude as those that occurred in the measurements made in the temple. Thus, we can consider an average heat capacity in the hand of 5.3 ± 0.4 J/K with an uncertainty of 8%. However, the equivalent thermal resistance obtained in these two measurements is clearly different, so that on the first measured day, with less heat dissipation, the equivalent thermal resistance is 35% greater than that obtained on the second day. With this discussion on the results obtained, we want to indicate that while the heat capacity in an area of the skin keeps its value in a range of ±8%, the thermal resistance can change greatly depending on the physical situation of the subject.

### 3.3. Discussion

The magnitude of equivalent thermal resistance and heat capacity exposed in this work is global and of non-specific values, and is associated with the measured 2 × 2 cm^2^ skin area. The coefficient of variation is 3.5% for heat capacity and 6% for equivalent thermal resistance. These indicators are similar to values obtained in other works [10,15,16]. To compare our values with those in the literature, it is necessary to consider the thermal penetration depth. In our case, this is 3–4 mm depending on the measured area. From this condition, the specific heat capacities and thermal conductivities obtained from our experimental results are consistent with the literature. For example, for a skin volume of 2 × 2 × 0.4 cm^3^, considering an average specific heat capacity of 3.40 Jg^−1^K^−1^ and an average density of 1.15 g/cm^3^ [36,37], we obtain an absolute heat capacity of 6.26 JK^−1^. For an average thermal conductivity of 0.30 Wm^−1^K^−1^ [36,37], the thermal resistance would be 33.3 KW^−1^. These values are similar to our results. 

The results obtained in this work have been obtained from experimental measurements made with a previously calibrated instrument. A more precise interpretation of the experimental results requires a better study of the skin’s thermal model. An increase in thermal penetration depth implies an increase in the heat capacity and thermal resistance of the zone. Other variables that affect the thermal properties are the physical state of the subject and the presence of lesions in the measurement zone. The physical state of the subject significantly affects the equivalent thermal resistance and, to a lesser extent, the heat capacity. 

## 4. Conclusions

In this work a calorimetric sensor has been used to measure, by applying it to the skin, the heat flow dissipated by the human body. A non-invasive method is proposed to determine the heat capacity and equivalent thermal resistance of the measured skin area. The modelling of the sensor is effective since it is capable of relating the signals measured by the sensor with the heat flow that passes through the sensor. A sensor calibration has been carried out to determine the invariant parameters of the model. A calculation method that allows one to reliably determine the heat flow and the thermal properties of the skin in the measured area has been proposed. We have verified that the heat capacity of the skin is a property whose value remains at a value that can vary by 8%. However, the equivalent thermal resistance of the skin has a greater variability due to its great dependence on the physical situation of the subject.

With this work, we show the ability of the sensor to determine in vivo the heat flow, the equivalent thermal resistance and the heat capacity of the skin using a non-invasive technique. These physical magnitudes show a great variability and depend on physiological and environmental factors. While this fact could be interpreted as a problem of uncertainty, it is an opportunity to advance the study of human physiology with new data provided by the sensor utilised in this work. The repeatability experiments made in each series show that the variability of the results responds more to changes in the subject and the environment than to changes in the device.

Manually clamping of the sensor on the skin is an inconvenience as it requires specialized personal. Currently, in order to determine the absolute values of the heat flow, it is necessary to transfer the sensor from its base to the skin, and this is carried out manually. However, for the determination of equivalent thermal resistance and heat capacity, it is not necessary for the sensor to be previously on the base; instead, it can be permanently placed on the skin by means of an adapted clamp. We are currently working on this possibility by thermally exciting the skin with sinusoidal thermal dissipations.

## Figures and Tables

**Figure 1 sensors-20-03431-f001:**
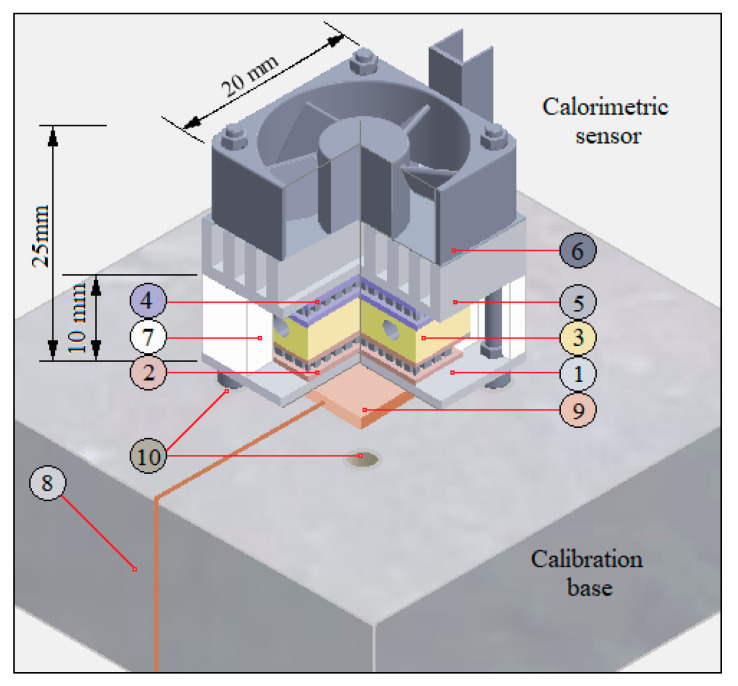
Diagram of the sensor in its calibration base. (1) aluminium plate, (2) measuring thermopile, (3) thermostat, (4) cooling thermopile, (5) aluminium heatsink, (6) fan, (7) thermal insulation, (8) calibration base (insulating material), (9) copper plate containing the calibration resistance, (10) magnets to hold the sensor in the base.

**Figure 2 sensors-20-03431-f002:**
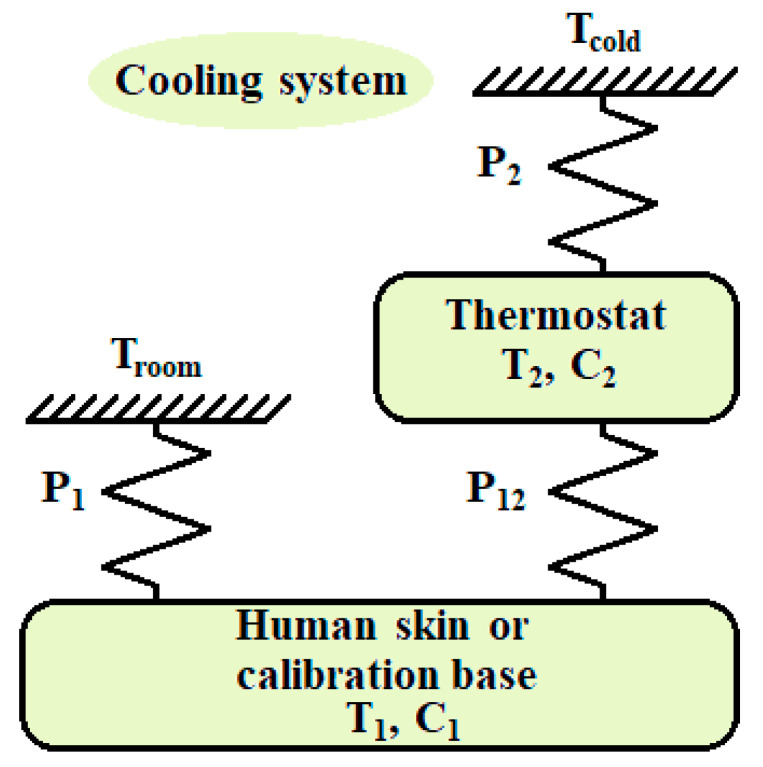
Calorimetric sensor model.

**Figure 3 sensors-20-03431-f003:**
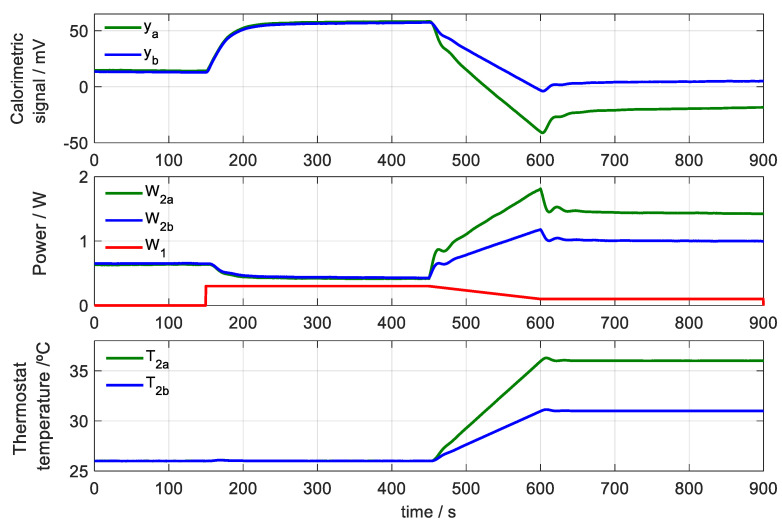
Experimental calibration measurements. Power dissipated in the base (*W*_1_), power dissipated in the thermostat (*W*_2*a*_ and *W*_2*b*_), calorimetric signal (*y_a_* and *y_b_*) and temperature of the thermostat (*T*_2*a*_ and *T*_2*b*_). The green curves represent the case of an increase of 10 K in temperature of the thermostat (from 26 to 36 °C), and the blue curves an increase of 5 K (from 26 to 31 °C). *T_room_* = 23.4 °C.

**Figure 4 sensors-20-03431-f004:**
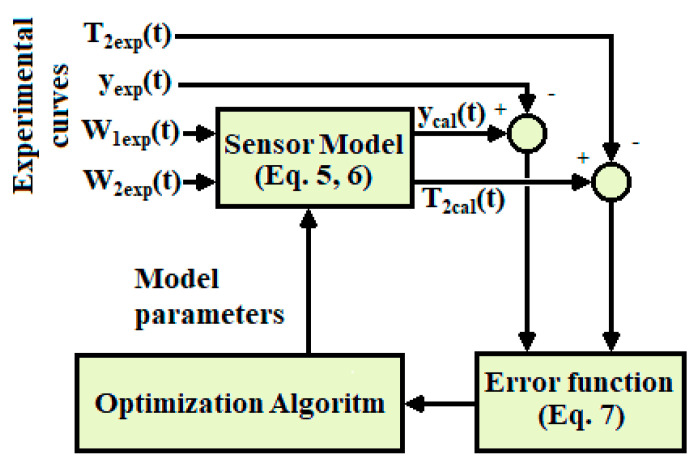
Model calculation procedure.

**Figure 5 sensors-20-03431-f005:**
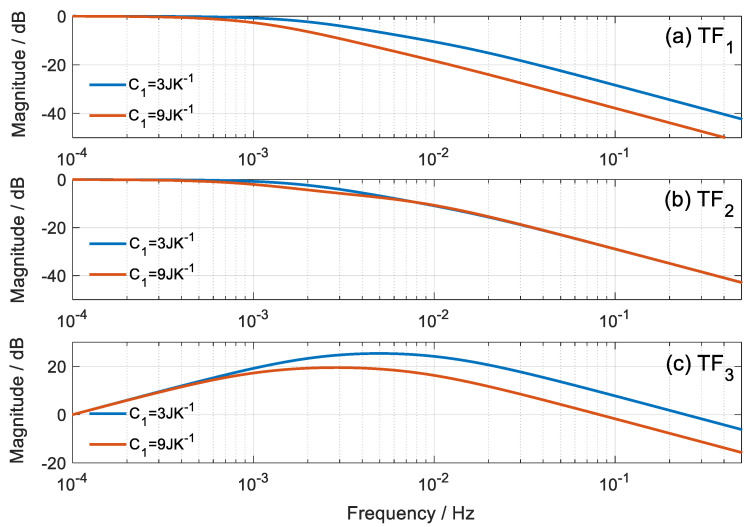
Bode representation of the magnitude of the *TF_i_* that relates the calorimetric signal to the inputs of the calorimetric system.

**Figure 6 sensors-20-03431-f006:**
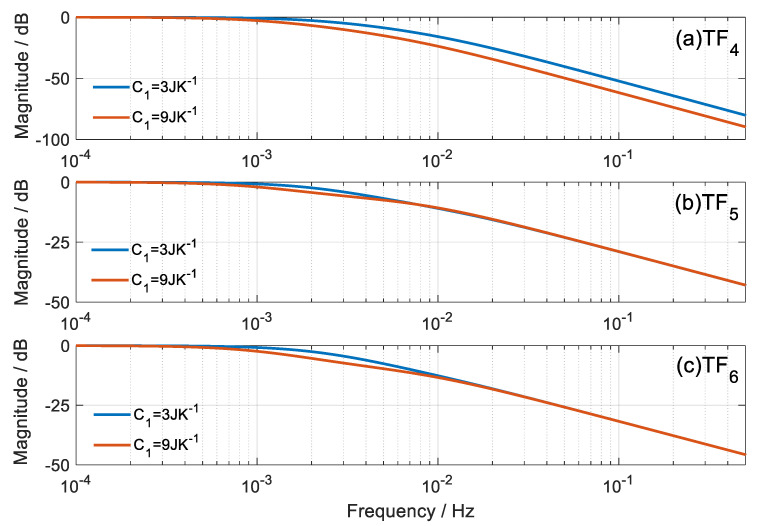
Bode representation of the magnitude of the *TF_i_* that relates the temperature of the thermostat with the inputs of the calorimetric system.

**Figure 7 sensors-20-03431-f007:**
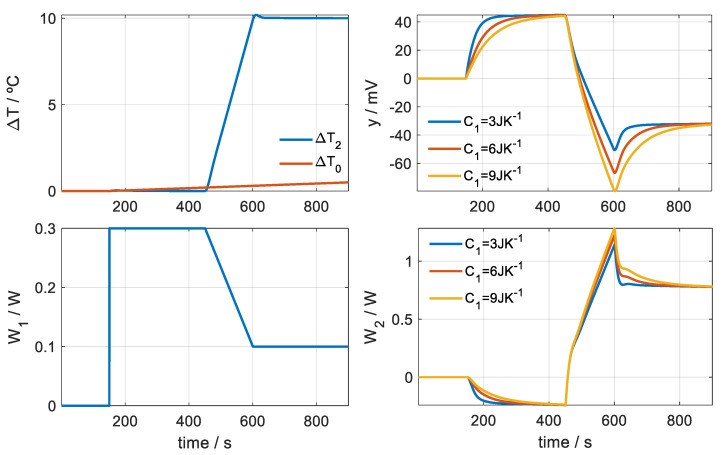
Simulations for the case of calibrations on a base that has different heat capacities. Curve baselines have been corrected.

**Figure 8 sensors-20-03431-f008:**
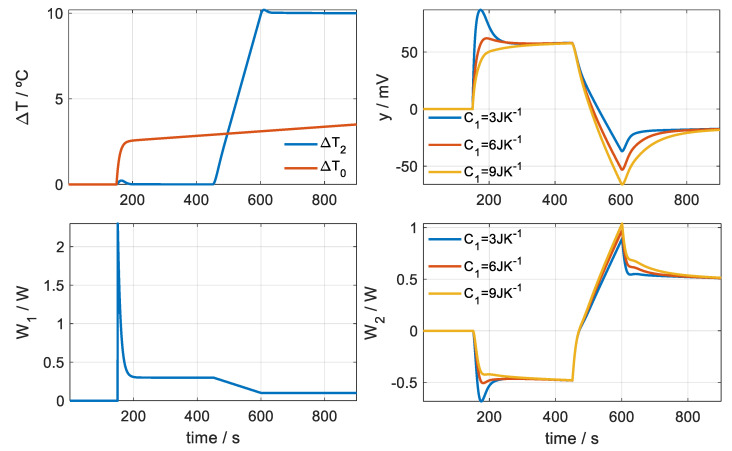
Simulation of a measurement on human skin for different heat capacities of the skin. Curve baselines have been corrected.

**Figure 9 sensors-20-03431-f009:**
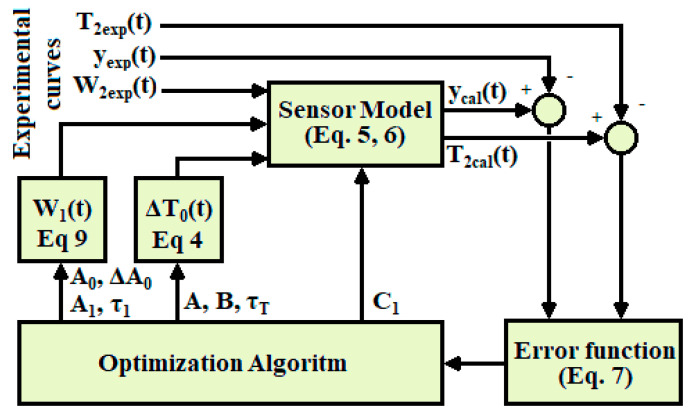
Heat capacity and heat flux calculation procedure.

**Figure 10 sensors-20-03431-f010:**
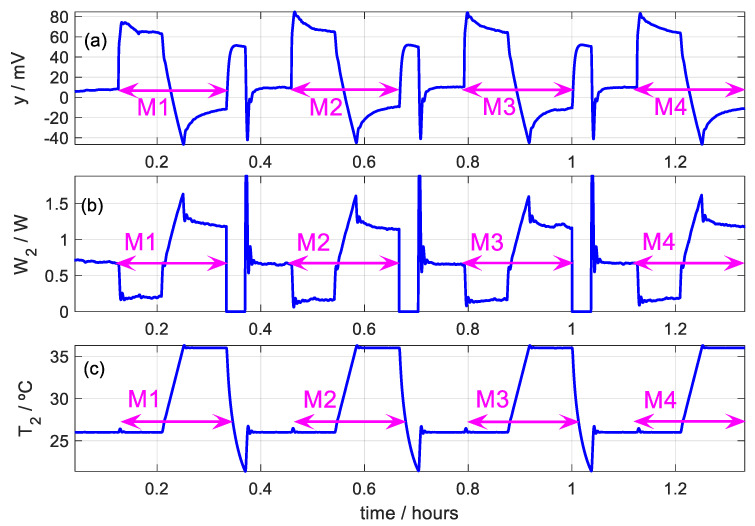
Series of four consecutive measurements (M1, M2, M3 and M4) performed on the temple of a healthy 28-year-old male subject (junior subject). (**a**) Calorimetric signal. (**b**) Power dissipated in the thermostat. (**c**) Thermostat temperature (*T_room_* = 22.6 ± 0.5 °C, 53% RH).

**Figure 11 sensors-20-03431-f011:**
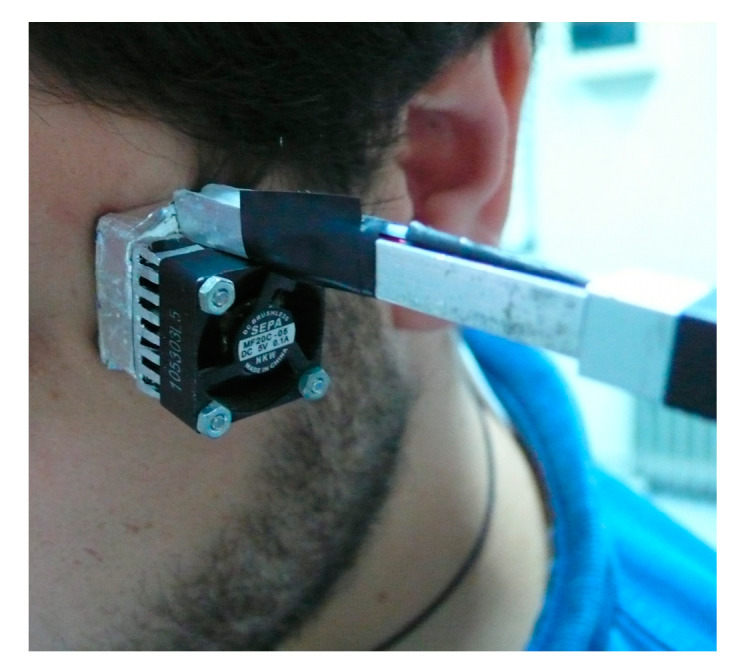
Sensor application to the skin.

**Figure 12 sensors-20-03431-f012:**
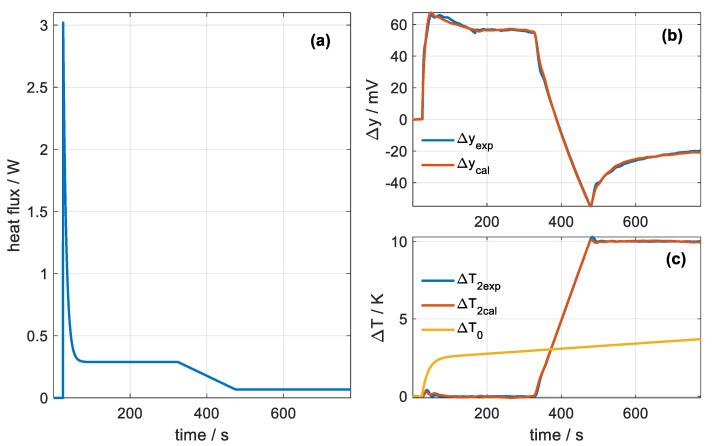
(**a**) Heat flow obtained from measurement M1 in Figure 10. (**b**) Adjustment of the calorimetric signal Δ*y*. (**c**) Thermostat temperature fitting Δ*T*_2_ and representation of Δ*T*_0_.

**Figure 13 sensors-20-03431-f013:**
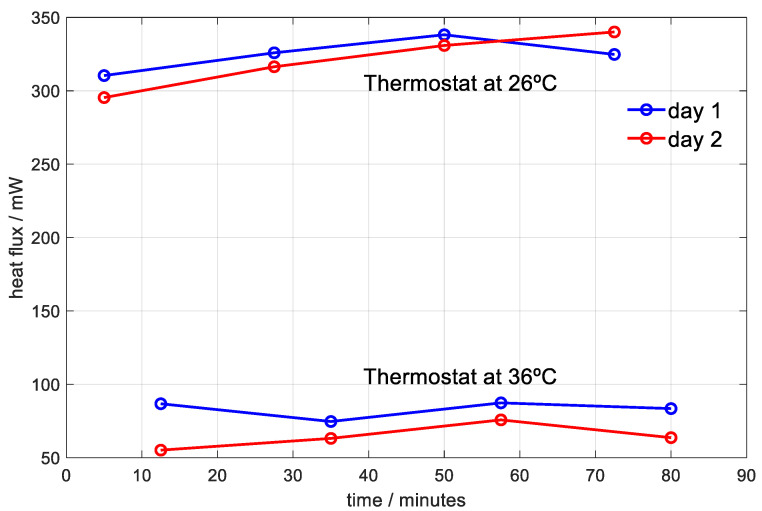
Heat flux transmitted from the temple to the sensor thermostat on different days and for different thermostat temperatures. Senior subject.

**Table 1 sensors-20-03431-t001:** Model parameters (Figure 2 and (Equations (5) and (6)) obtained in the calibration measurements.

Parameter	*C*_1_/JK^−1^	*C*_2_/JK^−1^	*P*_1_/mWK^−1^	*P*_2_/mWK^−1^	*P*_12_/mWK^−1^	*k*/mVK^−1^
Mean	3.00	3.98	33.38	64.83	96.07	19.02
SD	0.11	0.10	2.77	3.91	7.19	1.09
SD: Standard deviation. Number of measurements: 20. Maximum RMSE: *ε_y_* = 14 μV, *ε*_*T*2_ = 4 mK (Equation (7))						

**Table 2 sensors-20-03431-t002:** Results of the calculation on the simulated measurements with *C*_1_ = 3, 6 and 9 JK^−1^ (Figure 8).

	Heat Flux (Equation (9))				Model (Equation (5))	Δ*T*_0_ (Equation (4))			Errors (Equation (7))	
	*A*_0_/mW	*A*_1_/mW	*τ*_1_/s	Δ*A*_0_/mW	*C*_1_/JK^−1^	*A*/K	*B*/K	*τ_T_*	*ε_y_*/µV	*ε_T_*/mK
(1)	300.5	2125.9	8.4	−200.3	3.01	2.51	0.99	9.0	8.88	0.13
(2)	300.4	2125.5	8.4	−200.5	6.00	2.50	0.99	9.0	4.70	0.09
(3)	300.3	2123.6	8.4	−200.6	8.98	2.50	0.99	9.0	3.29	0.08

**Table 3 sensors-20-03431-t003:** Data of the subjects.

Subject	Gender	Age	Weight (kg)	Height (m)
Junior	Male	28	67	1.72
Senior	Male	62	71	1.65

**Table 4 sensors-20-03431-t004:** Results in four consecutive measurements performed on the temple (junior subject, Figure 9).

Measure	Heat Flux (Equation (9))				Model (Equation (5))	Δ*T*_0_ (Equation (4))			Errors (Equation (7))	
	*A*_0_/mW	*A*_1_/W	*τ*_1_/s	Δ*A*_0_/mW	*C*_1_/JK^−1^	*A*/K	*B*/K	*τ_T_*	*ε_y_*/µV	*ε_T_*/mK
M1	288	2.73	7.9	−221	5.69	2.46	1.23	16.6	29.7	1.47
M2	279	3.24	8.1	−216	5.82	2.68	1.60	9.00	33.5	2.03
M3	272	3.38	8.0	−203	5.96	2.80	1.05	9.00	34.7	3.78
M4	272	3.82	7.0	−214	6.17	2.86	1.11	9.00	29.1	2.55
Mean	278	3.29	7.8	−214	5.91	2.70	1.25	10.9		
SD	7.6	0.45	0.5	7.6	0.21	0.18	0.15	3.80		

**Table 5 sensors-20-03431-t005:** Results of measurements made on the temple in the senior subject on two different days (day 1: *T_room_* = 23.3 °C, day 2: *T_room_* = 23.5 °C).

Measure Day		Heat Flux (Equation (9))				Model (Equation (5))	Δ*T*_0_ (Equation (4))			Errors (Equation (7))	
		*A*_0_/mW	*A*_1_/W	*τ*_1_/s	Δ*A*_0_/mW	*C*_1_/JK^−1^	*A*/K	*B*/K	*τ* *_T_*	*ε_y_*/µV	*ε_T_*/mK
M1	1	310.4	3.03	7.3	−223.6	5.53	2.10	0.80	9.00	24.4	2.59
M2		325.9	4.89	5.3	−251.2	6.08	2.41	1.46	9.00	32.3	1.75
M3		338.2	3.21	7.4	−250.8	5.37	2.26	1.16	30.0	32.6	2.00
M4		324.8	3.55	7.0	−241.3	5.91	2.57	0.69	21.0	28.3	2.69
M1	2	295.4	2.98	7.5	−240.2	5.91	2.59	0.84	30.0	29.2	1.15
M2		316.4	2.82	9.1	−253.2	5.83	2.45	0.81	9.00	35.7	2.02
M3		330.9	3.73	6.6	−255.1	6.00	3.08	0.60	30.0	25.8	1.54
M4		340.1	3.29	8.0	−276.4	5.85	2.71	1.09	22.9	32.1	1.31

**Table 6 sensors-20-03431-t006:** Results of measurements made on the right hand in the senior subject in two different days (day 1: *T_room_* = 21.3 °C, day 2: *T_room_* = 21.8 °C).

Measure Day		Heat Flux (Equation (9))				Model (Equation (5))	Δ*T*_0_ (Equation (4))			Errors (Equation (7))	
		*A*_0_/mW	*A*_1_/W	*τ*_1_/s	Δ*A*_0_/mW	*C*_1_/JK^−1^	*A*/K	*B*/K	*τ_T_*	*ε_y_*/µV	*ε_T_*/mK
M1	3	15.5	−0.271	5.0	−202.0	5.08	0.61	0.49	9.30	42.3	1.60
M2		12.9	−0.037	17.9	−200.3	5.23	0.63	0.60	9.00	23.6	1.96
M3		26.1	0.055	10.7	−205.8	4.83	0.93	0.93	9.80	17.0	0.97
M1	4	295.5	2.114	6.9	−248.4	4.85	1.88	0.96	9.00	26.3	1.70
M2		300.4	3.217	6.4	−247.2	6.04	1.96	0.73	30.0	35.3	2.75
M3		284.1	2.335	7.7	−259.8	5.50	1.73	0.71	9.60	24.1	1.62
M4		246.9	1.935	8.3	−263.7	5.31	1.43	0.81	9.00	26.6	2.42

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
