# Peer review of "A Method to Determine Human Skin Heat Capacity Using a Non-Invasive Calorimetric Sensor"

_sensors, 2020, doi:10.3390/s20123431_

Round 1

Reviewer 1 Report

Several important issues arise form the reading of the manuscript.   In the introduction, usually authors provide a detailed description of the context of application and of the state-of-the-art instead of the functioning model.   Also due to the lack of the information about the latest literature, it is not clear if and how the proposed method is novel.   What are the practical application of such a method that requires calibration for each use and application time over the skin ranging from 5 to 15 minutes?   How the system is different from the one proposed in:  Socorro, F.; Rodríguez de Rivera, P.J.; Rodríguez de Rivera, M. Calorimetric minisensor for the localized measurement of surface heat dissipated from the human body. Sensors 2016, 16, 186?   Properly references must be provided for the equations, unless they are derived in this work (in this case, specify it).   A manual connection between the sensor and the human temple for a time greater than one hour is difficult to maintain. How this measurement can be considered reliable?   The numerical results of the heat capacity and equivalent thermal resistance are not sufficient to clearly highlight the novelty of the proposed method   Page 9, line 304 and 305: measurements simulated / simulation of measurements Are them measurements or simulations?   Minor comments: Check the template and in particular the font size  In several parts “y” is used instead of “and"   References: Most of the references are from the same authors of the papers. Different and updated references form other research groups help to describe the differences of the proposed method with state-of-the-art.

Author Response

We try to answer all your questions:

1) A new text has been included at the beginning of the introduction placing the work in its context. Lines 33-47 of the new text:

The study of the thermal dissipation of the human body is of great interest in multiple fields. In the air conditioning projects, it is necessary to know the dissipation of the occupants according to their activity. In the study of a subject's metabolism, thermal dissipation is determined indirectly, measuring the absorbed VO2 or VCO2 emitted by the subject [1]. In the field of human physiology, all available data and techniques are used. Contact and remote thermometry are irreplaceable tools. There are numerous publications that provide temperature data of different areas and organs of the human body [2]. Also of interest are the thermal properties (thermal conductivity and specific heat capacity) of different parts of the human body. Heat capacity is mainly determined from tissue sample analysis using techniques such as differential scanning calorimetry (DSC). These results are used for the study of pathologies [3-4] and the development of new bio-thermal models [5-7].

Sensors applied to the skin are of great importance as they are part of non-invasive diagnostic techniques [8]. New sensors applicable to the skin are currently being developed to determine its thermal conductivity [9] and also to determine the core temperature of the human body [10-11]. This work introduces a new application of a calorimetric sensor that allows to obtain the thermal properties of the skin. This calorimetric sensor has ….

2) Clarification of sensor calibration. Lines 115-117 of new text:

"The recalibration of the sensor shall be done when the sensor is disassembled for technical reasons or when it is necessary to verify its operation".

3) Reference has been included in equations 1 and 2 (lines 160 and 167). Moreover, a new variable ΔT0 included in this job. This implies a more complete model that significantly better fits experimental curves.

4) Since we built this prototype we have made numerous measurements in the human body studying the response obtained by the sensor and how to improve its functioning. We have verified that manual clamping of the sensor is an inconvenience. We include a new text at the end of the conclusions. Lines 468-474 of the new text:

"Manually clamping of the sensor on the skin is an inconvenience as it requires specialized personal. Currently, in order to determine the absolute values of the heat flow, it is necessary to transfer the sensor from its base to the skin and this is done manually. However, for the determination of equivalent thermal resistance and heat capacity it is not necessary for the sensor to be previously on the base, but can be permanently placed on the skin by means of an adapted clamp. We are currently working on this possibility by thermally exciting the skin with sinusoidal thermal dissipations."

5) A measurement of 12.5 minutes is sufficient to determine the thermal properties. In this work, with the intention of evaluating the uncertainty of the results, 4 replicates have been made in each measurement.

6) Correction on lines 321-322:

Old text: “… is verified from the measurements simulated in the...”

New text: “…is verified from the curves simulated in the...”

7) Font size and format have been checked.

Reviewer 2 Report

The authors report a methodology to measure thermal properties of skin by using a non-invasive calorimetric sensor. Systematically experimental studies on several subjects reveal the fundamental aspects of operation of the sensor and the underlying physics associate with the prediction of thermal properties of skin. The authors show convincingly that the sensor can measure the thermal resistance of skin in the measured area. These results are of interests and may create engineering opportunities to monitor skin thermal properties for skin disease diagnosis. The paper is well written and constructed. Several issues should be addressed before considering its publications.

  1. The intimate contact of the senor and skin is required to measure the temperature accurately. To have a conformal contact, a pressure must be applied on the sensor in experiments. Please comment on the effect of this pressure on the results.
  2. The sensor used in this paper is rigid and is not friendly to human skin. Recent advances of mechanics designs and materials have enabled the developments of epidermal/wearable electronics (e.g., Science 333, 838-843, 2011; ACS Applied Materials & Interfaces 11, 14340-46, 2019), which can measure skin temperature continuously and are mechanically invisible to skin. Some comments on the skin-like temperature sensors are expected.
  3. In conclusion section, in addition to the innovative aspects related to their work, limitations are expected and would help readers to understand the importance of the work.

Author Response

1) The application of the sensor on the skin is done with a normal pressure that is not painful for the subject. Any defect related to the contact between the sensor and the skin involves significant disturbances in the signals, so that abnormal measurements are easily detected.

2) Since we built this prototype we have made numerous measurements in the human body studying the response obtained by the sensor and how to improve its functioning. We have verified that manual clamping of the sensor is an inconvenience. We include a new text at the end of the conclusions. Lines 468-474 of the new text:

"Manually clamping of the sensor on the skin is an inconvenience as it requires specialized personal. Currently, in order to determine the absolute values of the heat flow, it is necessary to transfer the sensor from its base to the skin and this is done manually. However, for the determination of equivalent thermal resistance and heat capacity it is not necessary for the sensor to be previously on the base, but can be permanently placed on the skin by means of an adapted clamp. We are currently working on this possibility by thermally exciting the skin with sinusoidal thermal dissipations."

3) The use of new materials is not ruled out, although we are currently focused on performing a clinical validation of the sensor. We think that the determination of these two thermal parameters (heat capacity and equivalent thermal resistance) can be of great interest. As indicated in the last paragraph of the conclusions, we are looking for new procedures.

Reviewer 3 Report

This paper presents a non-invasive method for the heat flow estimation. A calorimetric sensor previously designed and developed by the same research team has been used to measure the heat flow dissipated by the human body. The method is proposed to determine the heat capacity and equivalent thermal resistance of the measured skin region. After a careful modelling analysis, the measuring system and the non-invasive method for the parameters estimation have been tested on volunteers.

The manuscript describes clearly and in detail the methods and the results. I consider this work suitable for publication in Sensors MDPI.

One comment about the introduction can allow further improving the paper.

Indeed, the introduction appears more as a discussion than as an introduction. The framework and the state-of-the-art is poorly described. I suggest rewriting this section, according to the following organization:

1) the state-of-the-art and application field

2) issues to be addressed

3) proposed solution(s)

4) brief description of the main characteristics of the proposed system/solution

Regarding the reference list, I have noted that 11 out 20 references are of the same team. Please remove the references less pertinent to this work. If possible and meaningful for the topic, I suggest also to expand the literature to be used as a reference.

Author Response

1) A new text has been included at the beginning of the introduction placing the work in its context. Lines 33-47 of the new text:

The study of the thermal dissipation of the human body is of great interest in multiple fields. In the air conditioning projects, it is necessary to know the dissipation of the occupants according to their activity. In the study of a subject's metabolism, thermal dissipation is determined indirectly, measuring the absorbed VO2 or VCO2 emitted by the subject [1]. In the field of human physiology, all available data and techniques are used. Contact and remote thermometry are irreplaceable tools. There are numerous publications that provide temperature data of different areas and organs of the human body [2]. Also of interest are the thermal properties (thermal conductivity and specific heat capacity) of different parts of the human body. Heat capacity is mainly determined from tissue sample analysis using techniques such as differential scanning calorimetry (DSC). These results are used for the study of pathologies [3-4] and the development of new bio-thermal models [5-7].

Sensors applied to the skin are of great importance as they are part of non-invasive diagnostic techniques [8]. New sensors applicable to the skin are currently being developed to determine its thermal conductivity [9] and also to determine the core temperature of the human body [10-11]. This work introduces a new application of a calorimetric sensor that allows to obtain the thermal properties of the skin. This calorimetric sensor has ….

2) References has been modified

Round 2

Reviewer 1 Report

The authors partially fulfilled my previous requests.
However, some comments still need a proper response, in the view of improving the final manuscript in order to be worth of publishing in Sensors.

The referenced literature has been increased, but a proper comparison on the difference of the actual solution implementation with the state-of-the-art still lacks.
How the other methods work? What are the lacks that can be overcame with this new method?
The main differences are related to the sensor, the arrangement, the post-processing of the signal, the duration of the measurement, the accuracy?
The other methods provide the thermal properties of the skin?

A plurality of contact and contactless sensors for temperature monitoring have been recently published on Sensors, few of them are:

DOI: 10.3390/s20030623
PVDF-TrFE-Based Stretchable Contact and Non-Contact Temperature Sensor for E-Skin Application. Sensors 2020, 20, 623.

DOI: 10.3390/s18010110
A Monolithic Multisensor Microchip with Complete On-Chip RF Front-End. Sensors 2018, 18, 110.

DOI: 10.3390/s18020645
Flexible, Stretchable Sensors for Wearable Health Monitoring: Sensing Mechanisms, Materials, Fabrication Strategies and Features. Sensors 2018, 18, 645.

The other concern is related to the numerical results of the heat capacity and equivalent thermal resistance that should be compared to other works to clearly highlight the novelty of the proposed method.

Author Response

1) New paragraph in the introduction (in blue in new text). Included new references.

New text: "Currently, new sensors with materials adaptable to the skin are being developed [13-16]. This new generation of sensors perform measurements on the skin with a thermal penetration depth of 0.15 mm (stratum corneum). However, the sensor object of this work provides macroscopic thermal values ​​with a thermal penetration depth of 3 - 4 mm. We believe these measurements with a higher thermal penetration depth are of interest and complement other thermal measurements."

2) Included section 3.3 before the conclusions (in blue in the new text).

3.3. Discussion.

The magnitude of equivalent thermal resistance and heat capacity exposed in this work are global and non-specific values, and are associated with the measured 2x2 cm2 skin area. The coefficient of variation is 3.5% for heat capacity and 6% for equivalent thermal resistance. These indicators are similar to values obtained in other works [10, 15, 16]. To compare our values with the literature ones, it is necessary to consider the thermal penetration depth. In our case is 3 - 4 mm depending on the measured area. From this condition, the specific heat capacities and thermal conductivities obtained from our experimental results are consistent with the literature. For example, for a skin volume of 2x2x0.4 cm3, considering an average specific heat capacity of 3.40 Jg-1K-1 and an average density of 1.15 g / cm3 [36, 37] we obtain an absolute heat capacity of 6.26 JK-1. For an average thermal conductivity of 0.30 Wm-1K-1 [36-37], the thermal resistance would be 33.3 KW-1. These values ​​are similar to our results.

The results obtained in this work have been obtained from experimental measurements made with a previously calibrated instrument. A more precise interpretation of the experimental results requires a better study of the skin thermal model. An increase of thermal penetration depth implies an increase in the heat capacity and thermal resistance of the zone. Other variables that affect to thermal properties are the physical state of the subject and the presence of lesions in the measurement zone. Physical state of the subject significantly affects the equivalent thermal resistance and, to a lesser extent, the heat capacity.

Thank for suggestions.